# SWEET: Weakly Supervised Person Name Extraction for Fighting Human Trafficking

**Javin Liu**[*]
University of Southern California
Mila – Quebec AI Institute

**Hao Yu**[*]
**Vidya Sujaya**[*]
**Pratheeksha Nair**
**Kellin Pelrine**
**Reihaneh Rabbany**
McGill University
Mila – Quebec AI Institute

## Abstract

In this work, we propose a weak supervision pipeline SWEET: *Supervise Weakly for Entity Extraction to fight Trafficking* for extracting person names from noisy escort advertisements. Our method combines the simplicity of rule-matching (through *antirules*, i.e., negated rules) and the generalizability of large language models fine-tuned on benchmark, domain-specific and synthetic datasets, treating them as weak labels. One of the major challenges in this domain is limited labeled data. SWEET addresses this by obtaining multiple weak labels through labeling functions and effectively aggregating them. SWEET outperforms the previous supervised SOTA method for this task by 9% F1 score on domain data and better generalizes to common benchmark datasets. Furthermore, we also release HTGEN, a synthetically generated dataset of escort advertisements (built using ChatGPT) to facilitate further research within the community.

## 1 Introduction

Over 6.3 million people worldwide are victims of forced sexual exploitation or human trafficking (HT) on any given day (International Labour Organization), and the majority of the victims have been advertised online (Polaris, 2021; MINOR, 2015) through escort websites (Rhodes, 2016). HT is an organized crime and traffickers tend to advertise multiple victims simultaneously. Hence, finding connections between online escort advertisements hints towards them being posted by the same individual and strongly indicates organized activity.

Several methods have previously taken the approach of uncovering connections between ads by looking for repeated phrases, phone numbers, locations, prices, service types, etc., in the text (Lee et al., 2021; Tong et al., 2021; Rabbany et al., 2018). Hence, efficient entity extractors must extract accurate and relevant information from ad text (Nagpal

et al., 2017; Li et al., 2022; Kejriwal et al., 2018; Kejriwal and Kapoor, 2019). However, this is very challenging because the text is often noisy, ungrammatical, and obscured. Information related to the names of individuals, services, locations, etc, often have letters replaced by symbols or emojis. The writing style constantly evolves and is made adversarial to avoid detection by online filters and moderators.

Based on discussions with domain experts, including criminologists, law enforcement, and other groups fighting trafficking, it was determined that extracting advertised names from the ads is particularly important. Traffickers tend to control, on average, four to six victims and often post advertisements on their behalf (Thorn, 2015). Hence, multiple people being advertised in the same or similar ads is a strong indicator of trafficking (Lee et al., 2021). Other entities such as locations, phone numbers and email addresses could also be informative. However, most websites have explicit fields for these entities which makes them more straightforward to extract. The same is not true for person names which are usually embedded inside the text of the ads and are much more complex to extract.

Traditional extraction methods, such as the embedding-based Flair (Akbik et al., 2018a) and Spacy NLP models, do not do well on this type of data because they are sensitive to noise and tend to decrease in performance as more noise is injected into the data (Li et al., 2022). NEAT (Li et al., 2022) stands as the current SOTA name extractor in this domain and uses an ad-hoc combination of a name dictionary, a rule dictionary, and a BERT-based disambiguation layer. Moreover, NEAT relies on a sample of manually labeled data to provide optimal results. This is a challenge because the relevant datasets for this task are typically curated in-house and not shared publicly due to their sensitive nature. Similar constraints also apply to methods such as crowd-sourcing.

---

[*]These authors contributed equally to this research.

To address these challenges, we propose a weakly supervised method SWEET (Supervise Weakly for Entity Extraction to fight Trafficking)[1], that efficiently extracts person names from escort ads and is independent of training labels. We define label-independence in the scope of our work as not requiring domain or task-specific labels. SWEET uses a novel combination of fine-tuned language models (elucidated in Section 3), antirules, and an HMM-based method of aggregating them. Our contributions are:

- **Novel**: We study the problem of name extraction from noisy text through the novel lens of weak supervision and propose a new method.

- **Effective**: The proposed method SWEET is very effective and outperforms the previous SOTA in person name extraction on domain data by 9% F1 score.

- **Label-efficient**: SWEET leverages *weak labels* that are easy and cheap to obtain. This is very useful in several real-world applications, particularly in the HT domain, where obtaining labels is expensive and challenging due to the sensitive nature of the data.

- **Improving reproducibility**: We also introduce HTGEN, a synthetically generated dataset of escort ads which, unlike the real-world datasets used in papers in this domain, can be published for further research and helps with reproducibility.

## 2 Background

In this section, we cover the background information and related works necessary to understand this paper's context better.

**Named Entity Recognition** Named Entity Recognition (NER) is the task of identifying elements in the text that correspond to high-level categories like PERSON, ORGANIZATION, LOCATION, DATES, and other concrete concepts that can be explicitly named. The NER task aims to extract all occurrences of such entities. In this paper, the focus is on human names because they are most associated with an HT victim and names are the most common entity type to appear in escort ads.

**Entity extraction from escort ads** This is an important task that involves identifying and extracting named entities from a given ad. Rule-based models typically rely on regular expressions, handcrafted rules, and gazetteer-based approaches. For person names, NEAT (Li et al., 2022) is the SOTA method that combines a gazetteer (person name dictionary) with a regular expression extractor (rule dictionary) and a RoBERTa (Liu et al., 2019) based disambiguation layer called HT-bert. Such rule-based systems were shown(Li et al., 2022) to have a better performance over statistical models (Devlin et al., 2018; Peters et al., 2018; Liu et al., 2019) on the noisy escort ads dataset. However, on benchmark NER datasets, NEAT performs much worse. This is a likely result of its rule-based components designed to capture patterns in escort ads. Rule-based approaches can also be complementary to statistical-based approaches. In other domains, (Ratinov and Roth, 2009) argued that the NER task relies highly on external knowledge and shows that a statistical model combined with a rule-based gazetteer match makes a better-performing hybrid model.

**Weak supervision** Since we consider multiple sources of weak labels for our task, we rely on a popular weak-supervision framework, Skweak, for label aggregation and learning. Skweak (Lison et al., 2021) is a Python-based weak supervision framework made specifically for NER tasks. It works together with the Spacy library (Honnibal et al., 2020), allowing users to create labeling functions (LFs) that label an input text token as a type of entity and facilitates downstream processing. Skweak provides a variety of different LF types: heuristics, gazetteers and document-level functions allowing us to easily combine weak signals from our proposed combination of rule based systems and LLMs. These make Skweak a better choice compared to other weak label aggregators such as Snorkel (Ratner et al., 2016). The labels obtained from these LFs are later aggregated using a Hidden Markov Model (HMM).

The HMM's states and observations correspond to the true labels and LF outputs respectively. Skweak relies on a majority vote strategy (where each LF is a voter) to get a predicted label. This is then used to calculate the HMM's initial transition and emission probabilities, which are then updated until convergence using the Baum-Welch algorithm. To account for possible dependencies

---

[1]Our code is available at https://github.com/ComplexData-MILA/HT-NER

between LFs, Skweak tempers the probability density (weight) of an LF depending on its redundancy. It decreases the weight of an LF that produces a label of type 'PERSON', for example, if it shows high recall with other LFs that also produce 'PERSON' labels (high overlap in predictions). Post aggregation, a separate classifier model can be trained on the HMM's aggregated results and used directly on other datasets without requiring the LFs or the HMM.

**Large Language Models**   Recent works (Gilardi et al., 2023; Huang et al., 2023a) have shown that ChatGPT can be used as a data annotator and addresses ethical concerns regarding using human labelers on datasets that contain sensitive information. We experimented with using ChatGPT to extract person names from ads and used this labeled set to fine-tune BERT-based models as LFs in our framework.

We also experimented with using ChatGPT for generating escort ads to create an additional domain dataset. It has been shown that LLMs like ChatGPT exhibit stochastic behaviours (Bender et al., 2021) and are susceptible to biases from the real world internet data it was trained on. Recently, several works (Li et al., 2023) (Liu et al., 2023) have demonstrated effective jailbreak techniques that bypass the content filters that are imposed on ChatGPT to limit the level of bias and toxic language output by ChatGPT. (Carlini et al., 2021) could also extract training data directly from GPT LLMs.

## 3   Methodology

In this paper, we introduce a hybrid weakly supervised Skweak-based model that uses both rule-based approaches (such as "antirules" detailed below) and statistical-based approaches (such as DeBERTa and RoBERTa models trained on different NER datasets) as labeling functions and combine them into a single, more effective predictor. Our methodology pipeline is in Figure 1.

The proposed weak supervision pipeline in **SWEET** consists of 2 main types of LFs:

1. **Antirules**: Rules that determine entities as *not* person-names. The top X% most frequent words of the dataset (X $\in \{10, 20, 30, 40, 50\}$) are annotated as "NOT_NAME", resulting in 5 antirule LFs in total. The antirules help counteract possible noise.

2. **Fine-tuned Models**: We fine-tune base versions of RoBERTa (Liu et al., 2019) and DeBERTav3 (He et al., 2021) for the task of Named Entity Recognition on six different datasets, namely: the train splits of CoNLL2003 (Tjong Kim Sang and De Meulder, 2003), WNUT2017 (Derczynski et al., 2017), Few-NERD (Ding et al., 2021) (Only Level 1), WikiNER (English) (Nothman et al., 2012), a domain dataset labelled by ChatGPT called HTUNSUP, and a domain dataset generated and labeled by ChatGPT called HTGEN. The details and labeling method of HTUNSUP are provided in Section 4.6 and Appendix A, respectively. In total, 12 fine-tuned models (one per dataset and model type) act as LFs in **SWEET**. Any word predicted by a fine-tuned model, that matches a white-space-separated word in the ad text, gets annotated as a "PERSON_NAME" according to that LF.

Using the Skweak framework, we fit an HMM on the annotated train set and apply the fitted model on the annotations to obtain a final set of aggregated labels. The HMM's initial parameters are obtained using a majority voter that treats each LF as a voter of equal weight to get a predicted label.

We also experimented with using the three components of previous SOTA NEAT (Li et al., 2022) as LFs but ultimately found it did not boost performance (see NEAT (Weakly Supervised) in Table 3). Specifically, we considered:

1. **Context Rules**: extract person names based on part-of-speech (POS) tags and common phrases that include names. The rules consist of regular expressions that work to capture the most common contexts where names are seen. (eg. I'm NNP, My name is NNP, You can call me NNP).

2. **Name Dictionary**: common names with confidence scores tuned based on frequency in in-domain data. These names are used in gazetteer matching, where the names are used as the gazetteer and the pipeline, given a word in a sentence, will check if it is a word in this dictionary/gazetteer.

3. **Word Disambiguation**: filters results of the weighted output of rule and dictionary extractor with the help of a DeBERTaV3 model that was fine-tuned for the task of masked language modeling on a domain dataset (Li et al., 2022).

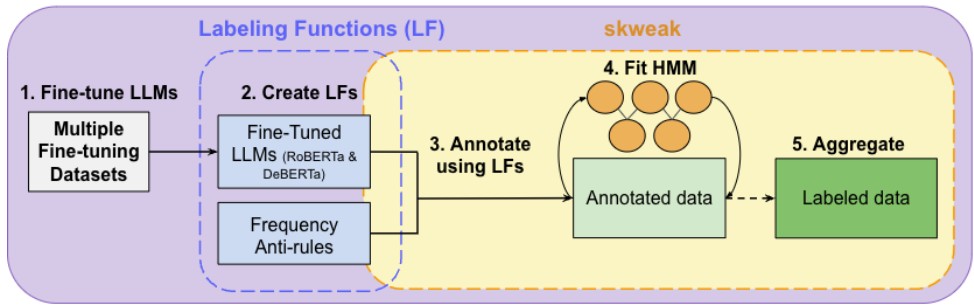

Figure 1: SWEET Architecture

During the training of the DeBERTaV3 model, individual words are masked and the task for the model is to predict these masked words. A word's context is passed to the fill mask, and the number of correct predictions (the number of predictions by the fill mask that is in the name dictionary) is divided by the total number of predictions by the fill mask to get a confidence score for the disambiguation.

## 4 Datasets

To fine-tune **SWEET** LFs, we used 4 open-source datasets, namely the training sets of CoNLL2003, WNUT2017, FewNERD-L1, and WikiNER-en, and 2 escort ad datasets, namely HTUNSUP and HTGEN. To evaluate **SWEET**, we used the test sets of CoNLL2003, WNUT2017, BTC, Tweebank and an escort ad dataset called HTNAME. We also generated a domain dataset HTGEN whose train set was used for fine-tuning LFs and test set was used for evaluating **SWEET**. We provide the dataset statistics in Table 1, and the descriptions below.

### 4.1 CoNLL2003

CoNLL2003(Tjong Kim Sang and De Meulder, 2003) is a very popular baseline for evaluating the performance of different NLP systems and algorithms. Our experiments use CoNLL2003 data from the HuggingFace which is in English. For evaluation, we only consider the name annotations (i.e, B-PER and I-PER) and ignore other entity classes.

### 4.2 WNUT2017

WNUT2017(Derczynski et al., 2017) consists of user-generated text and contains many examples of informal language, abbreviations, misspellings, and other noisy characteristics. Due to this, models tend to have lower recall values on WNUT2017.

Similar to CoNLL2003, we use only the B-PER and I-PER classes for evaluation.

### 4.3 Broad Twitter Corpus (BTC)

The Broad Twitter Corpus (BTC) (Derczynski et al., 2016) dataset is a large, diverse, and high-quality annotated corpus, created for the development and evaluation of NER in social media. BTC includes tweets from different regions, time periods, and types of Twitter users. It was annotated by a combination of NLP experts and crowd workers, ensuring both high quality and crowd recall.

### 4.4 Tweebank

Tweebank(Jiang et al., 2022) was developed to address the challenges of analyzing social media data, specifically Twitter messages through NER and syntactic parsing. The researchers created a new English NER corpus called Tweebank-NER based on Tweebank and annotated it using Amazon Mechanical Turk.

### 4.5 HTNAME

This is a modified version of the HT1k dataset from (Li et al., 2022) consisting of 995 escort advertisements in English and French, where we rectified three main types of labeling errors.

1. *Parsing issues* - in cases where the first name and last name were joined as a single string (FirstNameLastName) were not correctly annotated as "FirstName, LastName".
2. *Apostrophe issue* - cases where an extra s is added to the name. Examples like 'Mizz Mercede's parlour' was wrongly annotated as 'Mercedes' (the $s$ should not be included as the name).
3. *Blank labels* - a significant number of examples were missing true labels. We manually added the missing labels and reintroduce this dataset as HTNAME.

| Dataset | Train Size | Test Size | Total Size | Batch Size |
|---------|-----------|-----------|-----------|-----------|
| CoNLL2003 | 14041 | 3453 | 17494 | 128 |
| WNUT2017 | 3394 | 1287 | 4681 | 128 |
| FewNERD-L1 | 131767 | 18824 | 150591 | 32 |
| WikiNER-en | 129907 | 14435 | 144342 | 128 |
| BTC | - | 2002 | - | - |
| Tweebank | - | 1201 | - | - |
| HTNAME | - | 995 | 995 | - |
| HTUNSUP | 6160 | - | 6160 | 32 |
| HTGEN | 9424 | 818 | 10242 | 32 |

Table 1: Statistics of the datasets used. The grayed-out numbers indicate that the test sets are not used in our experiments.

## 4.6 HTUNSUP

HTUNSUP was gathered by us from private escort websites and has around 6000 ads. Multiple recent works have shown that ChatGPT's strong performance on text related tasks makes it a strong candidate for automatic labeling of certain types of data (Huang et al., 2023b; Mei et al., 2023; Zhu et al., 2023). Additionally, due to its outstanding performance on HTNAME (Table 2), we use it to generate pseudo labels for HTUNSUP, which we treat as ground-truth.

**Weak Labeling of HTUNSUP**   We used the ChatGPT API (*gpt-3.5-turbo*) to extract person-name entities from ads. The average token count including the prompt, ad and response was 263. Due to content moderation and dialogue optimization, on encountering improper or explicit text (as in the case of escort ads), ChatGPT responds incoherently and for such cases, we return "no entity detected".

The prompt used was: "*I want you to act as a natural and no-bias labeler, extract human names and location or address and social media link or tag in the format 'Names: \nLocations: \nSocial: '. If multiple entities exist, separate them by |. If no entity exists, say N. Your words should extract from the given text, can't add/modify any other words, should be as short as possible, remember don't include the phone number. For one name, it should be less than 3 words.*"

After post-processing and removing certain rare, high-frequency wrong predictions manually, we obtain HTUNSUP with high-quality pseudo labels and use it for fine-tuning BERT models.

| F1 | Precision | Recall |
|----|-----------|--------|
| 0.90 | 0.89 | 0.91 |

Table 2: Performance of ChatGPT for name extraction on HTNAME

## 4.7 HTGEN: Generated escort ads dataset

We explore ChatGPT's ability to generate escort ads with the motivation of developing a shareable domain specific dataset for furthering research in this field[2].

We used role playing to convince ChatGPT (API with GPT-3.5-turbo-0301) that it is a researcher studying patterns in escort ads. This bypasses the filters meant to reduce inappropriate outputs, misuse, and generation of text found in such ads. We followed up with a description of the entities in the ads and some of the patterns unique to the HT domain. (Bonifacio, 2023) had shown that providing the context before every question often helped improve the output quality. We also provided a sample conversation between a user (who provides the prompt) and ChatGPT (providing the required generated response) from three examples taken from our private HTNAME dataset. We scrambled phone numbers and other sensitive data for privacy reasons. In this manner, we generated 10,000 synthetic escort ads called HTGEN and used them to fine-tune a BERT model. At the time of writing, it costs $0.002 per 1000 tokens, and for 10,000 ads, it costs approximately $4.40 USD. This is significantly cheaper than hiring manual labelers.

We provide examples of ads from HTNAME and HTGEN in appendix D but note that some contain explicit language. The ads generated by ChatGPT are pretty similar to those in HTNAME, and ChatGPT was successfully able to replicate the patterns and noise of the HT domain.

## 5 Experimental Setup

### 5.1 Baselines

We compare SWEET with eight baselines, including off-the-shelf libraries, BERT-based models, domain-specific NER methods and a simple majority vote for aggregating weak labels as opposed to Skweak, ordered by their appearance in our result tables.

- **Spacy**[3] – A popular open-source Python library for natural language processing tasks. The Spacy NER package uses statistical models, neural network architectures and deep learning techniques trained on large amounts of data.

---

[2] The exact prompts used and the code for the dataset generation can be provided upon request.

[3] https://spacy.io/usage/linguistic-features#named-entities

- **TwitterNER** (Mishra and Diesner, 2016) – semi-supervised approach that combines text cluster information and word embeddings and uses a CRF for NER
- **LUKE** (Yamada et al., 2020) – transformer based model that learns contextualized embeddings, fine-tuned on TACRED dataset (Zhang et al., 2017)
- **ELMo** (Peters et al., 2018) – model that learns contextualized word representations, tuned for NER
- **Flair** (Akbik et al., 2018b) – trained character language model that learns contextualized string embeddings, tuned for NER
- **NEAT** (Li et al., 2022) – previous SOTA for person name extraction from escort ads
- **NEAT (Weakly Supervised)** – a variation of NEAT where each individual component is treated as an LF in our weak supervision pipeline
- **Majority Vote** – a simple strategy of choosing the label based on consensus of majority of the weak labels, competitor to Skweak.

## 5.2 Evaluation Settings

We compare the performance of **SWEET** with its variants and the baselines using the average F1 classification score on 5-folds of the test set. For each run, 1 fold acted as the test set while the remaining were used to fit the HMM. Since we only consider the 'PERSON' class, we adopt the following evaluation metrics to better fit our use case.

- *Word-level matching*: both ground truth and prediction are split into words (separated by a space) and each word is treated as a separate entity.
- *Lowercase*: both the ground truth and predictions are converted to lowercase.
- *Exact Match*: since the entity matching is on the word level, we consider strict match as a True Positive sample.

## 5.3 Results

Tables 3 and 4 summarize the results of our experiments for the general and HT domain datasets respectively.

**Overall performance of SWEET:** **SWEET** obtains the highest F1 score and recall on 4 out of 6 datasets, and beats NEAT on all datasets. On the HT datasets, **SWEET** performs the best. On general noisy datasets, WNUT2017 and BTC, **SWEET** similarly performs the best. On CoNLL2003, Flair

and ELMo achieve higher scores than **SWEET**, but **SWEET** is the next best performing model. On Tweebank, TwitterNER, designed specifically for tweets, is the best performing method. Please note that on re-evaluating NEAT on HTNAME, we found an increase in the strict F1 score from the previous results (0.76 from Table 2 in Li et al. (2022) becomes 0.78), and use this stronger performance as the baseline, as reported in Table 4.

**Comparison of SWEET with NEAT:** When compared with NEAT on the domain datasets, **SWEET** increases F1 score and recall by 9% and 18% respectively whilst maintaining precision. Furthermore, **SWEET** significantly surpasses NEAT on CoNLL2003, WNUT2017, BTC and Tweebank. This may be attributed to signals from the fine-tuned DeBERTa and RoBERTa LFs as well as false positive reduction due to the antirules.

**Performance of SWEET v.s. individual LFs:** In Table 5, we report the performance of individual labeling functions. When compared to the performance of **SWEET** in Table 4, we can see that **SWEET** successfully combines different fine-tuned models resulting in an aggregated model that outperforms these individual models, even those fine-tuned on domain data (HTUNSUP and HTGEN). **SWEET**'s recall and precision scores being higher than all individual fine-tuned LFs (except for precision on HTNAME which is on-par with HTUNSUP), shows the success of weak supervision in using weaker sources (LFs) to produce a stronger final model. The weak supervision methodology however can impact the performance significantly as we can see in Table 3 and Table 4, when comparing a simple majority vote performance with the proposed **SWEET**. The majority vote baseline still performs relatively well, and shows strong recall, however, **SWEET** performs significantly better in 5 out of 6 datasets.

**Ablation Study** Table 6 shows the results of ablation experiments of **SWEET**, focused on the HT datasets. Firstly, removing antirules decrease precision, showing that they are helpful in informing the model on what is not a name. Meanwhile, we see a mixed effect of domain LFs in the model. In row 3, we observe the highest precision and F1 scores on HTName with the removal of in-domain LFs. On HTGen, we instead see a significant decrease, but note that it is still higher than other baselines in Table 4. In the last rows, we observe that using only

| Method | CoNLL2003 | | | WNUT2017 | | | BTC | | | Tweebank | | |
|---|---|---|---|---|---|---|---|---|---|---|---|---|
| | F1 | Prec | Rec | F1 | Prec | Rec | F1 | Prec | Rec | F1 | Prec | Rec |
| spaCy (Honnibal et al., 2020) | .64 ± .04 | .66 ± .04 | .55 ± .04 | .21 ± .07 | .14 ± .06 | .44 ± .06 | .15 ± .01 | .44 ± .01 | .09 ± .01 | .19 ± .04 | .24 ± .05 | .18 ± .05 |
| TwitterNER (Mishra and Diesner, 2016) | .68 ± .05 | .91 ± .05 | .55 ± .05 | .61 ± .09 | .84 ± .10 | .57 ± .10 | .32 ± .05 | .86 ± .04 | .26 ± .04 | .67± .04 | .87± .05 | .57± .05 |
| LUKE (Yamada et al., 2020) | .31 ± .11 | .89 ± .09 | .19 ± .09 | .55 ± .07 | .67 ± .05 | .44 ± .05 | .47 ± .01 | .92 ± .01 | .31 ± .01 | .37 ± .07 | .76 ± .06 | .33 ± .06 |
| ELMo (Peters et al., 2018) | .96 ± .02 | .95 ± .02 | .99 ± .02 | .59 ± .15 | .72 ± .18 | .37 ± .18 | .41 ± .06 | .83 ± .04 | .35 ± .18 | .71± .01 | .74± .01 | .68 ± .01 |
| Flair (Akbik et al., 2019) | .98 ± .02 | .97 ± .02 | 1.0 ± .02 | .60 ± .15 | .79 ± .18 | .34 ± .18 | .35 ± .06 | .86 ± .05 | .31 ± .05 | .63± .07 | .76 ± .09 | .62 ± .09 |
| NEAT (Original) (Li et al., 2022) | .17 ± .07 | .43 ± .05 | .07 ± .05 | .22 ± .06 | .47 ± .04 | .16 ± .04 | .11 ± .02 | .46± .01 | .07 ± .01 | .24 ± .05 | .42 ± .04 | .17 ± .04 |
| NEAT (Weakly Supervised) | .16 ± .07 | .42 ± .05 | .07 ± .05 | .22 ± .06 | .47 ± .04 | .16 ± .04 | .11 ± .02 | .46 ± .01 | .07 ± .01 | .24 ± .05 | .42 ± .04 | .17 ± .04 |
| Majority vote | .83 ± .06 | .74 ± .02 | .98 ± .02 | .65 ± .04 | .53 ± .06 | .90 ± .06 | .64 ± .03 | .76 ± .03 | .60 ± .03 | .53 ± .06 | .37 ± .04 | .88 ± .04 |
| SWEET −Domain Data | .86 ± .05 | .79 ± .02 | .97 ± .02 | .69 ± .06 | .61 ± .06 | .82 ± .06 | .63 ± .03 | .86 ± .04 | .56 ± .04 | .58 ± .06 | .43 ± .04 | .85 ± .04 |
| SWEET | .86 ± .05 | .79 ± .03 | .98 ± .03 | .68 ± .04 | .58 ± .07 | .83 ± .07 | .64 ± .03 | .84 ± .04 | .57 ± .04 | .58 ± .06 | .43 ± .04 | .86 ± .04 |

Table 3: Word-level strict match results on 5-fold open source test sets: F1 score, Precision, Recall of SWEET and baselines. NEAT (Weakly Supervised) refers to the use of NEAT (Li et al., 2022) components as labeling functions in skweak.

| Method | HTNAME | | | HTGEN | | |
|---|---|---|---|---|---|---|
| | F1 | Prec | Rec | F1 | Prec | Rec |
| spaCy (Honnibal et al., 2020) | .27 ± .03 | .18 ± .02 | .51± .02 | .47 ± .04 | .50 ± .03 | .43 ± .03 |
| TwitterNER (Mishra and Diesner, 2016) | .56 ± .04 | 75. ± .04 | 52.± .04 | .70 ± .02 | .70 ± .03 | .67 ± .03 |
| LUKE (Yamada et al., 2020) | .63 ± .03 | .85 ± .04 | .51± .04 | .68 ± .03 | .84 ± .02 | .56 ± .02 |
| ELMo (Peters et al., 2018) | .51 ± .02 | .56 ± .02 | .46± .02 | .69 ± .05 | .61 ± .06 | .74 ± .06 |
| Flair (Akbik et al., 2019) | .45 ± .02 | .73 ± .02 | .32± .02 | .63 ± .04 | .83 ± .04 | .49 ± .04 |
| NEAT (Original) (Li et al., 2022) | .78 ± .04 | .83 ± .05 | .74± .03 | .71 ± .01 | .63 ± .03 | .79 ± .03 |
| NEAT (Weakly Supervised) | .79 ± .02 | .80 ± .02 | .77± .02 | .71 ± .01 | .64 ± .02 | .78 ± .02 |
| Majority vote | .73 ± .02 | .59 ± .01 | .95 ± .01 | .74 ± .02 | .65 ± .03 | .85 ± .03 |
| SWEET −Domain Data | .88 ± .01 | .85 ± .01 | .92 ± .01 | .75 ± .02 | .71 ± .03 | .78 ± .03 |
| SWEET | .87 ± .01 | .83 ± .01 | .92 ± .01 | .81 ± .02 | .76 ± .03 | .84 ± .03 |

Table 4: Word-level strict match results on 5-fold HT test sets: F1 score, Precision, Recall of SWEET and baselines. NEAT (Weakly Supervised) refers to the use of NEAT (Li et al., 2022) components as labeling functions in skweak.

| Model | Fine-tuning Dataset | F1 | Precision | Recall |
|---|---|---|---|---|
| DeBERTa-v3-base | HTUNSUP | .67 ± .03 | .71 ± .02 | .62 ± .02 |
| | HTGEN | .68 ± .01 | .71 ± .02 | .67 ± .02 |
| | CoNLL2003 | .67 ± .02 | .67 ± .03 | .69 ± .03 |
| | Few-NERD-L1 | .57 ± .03 | .80 ± .03 | .43 ± .03 |
| | WikiNER-en | .52 ± .01 | .48 ± .02 | .54 ± .02 |
| | WNUT2017 | .70 ± .02 | .71 ± .02 | .72 ± .02 |
| RoBERTa-base | HTUNSUP | .82 ± .02 | .84 ± .03 | .83 ± .03 |
| | HTGEN | .72 ± .02 | .81 ± .02 | .66 ± .02 |
| | CoNLL2003 | .72 ± .02 | .68 ± .03 | .77 ± .03 |
| | Few-NERD-L1 | .68 ± .02 | .81 ± .02 | .59 ± .02 |
| | WikiNER-en | .49 ± .03 | .43 ± .03 | .56 ± .03 |
| | WNUT2017 | .68 ± .02 | .73 ± .03 | .66 ± .03 |

Table 5: Fine-tuned model LFs on HTNAME evaluated on 5-fold test sets. All these LFs have a lower performance compared to the aggregated model SWEET, reported in Table 4.

domain LFs yields similar F1 scores with SWEET, but compromises recall for precision. We note that without using any domain data (second to last row of Table 4), SWEET is able to achieve the same F1 score as a setup that uses only domain data.

We also observe that removing HTUNSUP LFs results in an F1 score decrease only on HTNAME. Moreover, these LFs do as well on their own (last four rows in Table 6), although they are still lower than SWEET on HTNAME, showing that the aggregation of LFs fine-tuned on diverse datasets is a key component of SWEET.

# 6 Discussion

## 6.1 Effect of using ChatGPT

Although ChatGPT is the best performing model on HTNAME (Table 2), there are concerns regarding its use on escort ads. First, there is some financial cost associated with employing ChatGPT on large datasets of escort ads which limits its accessibility whereas our method is open-source and completely free to use. Second, the stability of the OpenAI API output is a concern as discussed in (Chen et al., 2023). While this does not hurt our method much where one only needs to get good performance a single time (to provide weak labels on a fixed dataset), and one could consider even the more powerful GPT-4, it means it would be unreliable to build a system for real-world applications based on ChatGPT alone. Finally, depending on the method used to access ChatGPT, there can be privacy concerns with this sensitive data. The intended use of our method is on large scale real-time ads with sensitive information, and using ChatGPT, is not feasible due to cost, stability and privacy issues mentioned above.

Moreover, although ChatGPT is a valuable ingredient in SWEET, our ablations directly show that it is not sufficient. In the target domain our full system achieves 87% F1 (Table 4). Simply apply-

| Method | HTName | | | HTGen | | |
|---|---|---|---|---|---|---|
| | F1 | Prec | Rec | F1 | Prec | Rec |
| All LFs − HTGEN LFs | .88 ± .01 | .85 ± .01 | .92 ± .01 | .81 ± .02 | .76 ± .03 | .84 ± .03 |
| All LFs − HTUNSUP LFs | .86 ± .02 | .82 ± .02 | .91 ± .02 | .82 ± .02 | .77 ± .03 | .84 ± .03 |
| All LFs − HTGEN LFs− HTUNSUP LFs (*Domain Data*) | .88 ± .01 | .85 ± .01 | .92 ± .01 | .75 ± .02 | .71 ± .03 | .78 ± .03 |
| All LFs −*Antirules* | .85 ± .01 | .79 ± .01 | .92 ± .01 | .81 ± .02 | .75 ± .03 | .84 ± .03 |
| All LFs −*Antirules*− HTGEN LFs | .85 ± .01 | .79 ± .01 | .93 ± .01 | .81 ± .02 | .75 ± .03 | .84 ± .03 |
| All LFs −*Antirules*− HTUNSUP LFs | .83 ± .02 | .76 ± .01 | .92 ± .01 | .81 ± .02 | .76 ± .03 | .84 ± .03 |
| All LFs −*Antirules*− HTGEN LFs − HTUNSUP LFs | .82 ± .02 | .76 ± .02 | .92 ± .02 | .81 ± .01 | .76 ± .03 | .84 ± .03 |
| Only HTGEN LFs | .76 ± .01 | .76 ± .02 | .76 ± .02 | .81 ± .02 | .79 ± .03 | .81 ± .03 |
| Only HTGEN LFs +*Antirules* | .86 ± .01 | .82 ± .01 | .92 ± .01 | .75 ± .04 | .82 ± .05 | .64 ± .05 |
| Only HTUNSUP LFs | .87 ± .01 | .88 ± .01 | .88 ± .01 | .81 ± .03 | .77 ± .04 | .81 ± .04 |
| Only HTUNSUP LFs +*Antirules* | .87 ± .01 | .92 ± .02 | .85 ± .02 | .76 ± .04 | .83 ± .05 | .66 ± .05 |
| Only HTGEN LFs + HTUNSUP LFs | .85 ± .01 | .85 ± .02 | .87 ± .02 | .82 ± .03 | .78 ± .03 | .81 ± .03 |
| Only HTGEN LFs + HTUNSUP LFs +*Antirules* | .86 ± .02 | .86 ± .01 | .86 ± .01 | .82 ± .03 | .79 ± .04 | .81 ± .04 |

Table 6: Ablation results of SWEET. −*Antirules* indicates removal of the antirule LFs component of SWEET. HTGEN LFs or HTUNSUP LFs indicates LMs fine-tuned on HTGEN or HTUNSUP respectively.

ing language models trained on ChatGPT-labeled data achieves 82% at best (Table 5), significantly lower. Conversely, our approach without using any ChatGPT-labeled data maintains 88% F1 (Table 6, "SWEET− DomainData").

## 6.2 Label Efficiency

Our pipeline eliminates the need for human labelers as it is independent of training labels for the task at hand, making use of existing open-source labeled data for fine-tuning language models instead. This is advantageous over other label-dependent NER models, especially in the HT domain where data labeling is costly and time-consuming. Being label independent is also beneficial in terms of ethics and privacy as the HT domain contains many examples of private information like names, locations, social media tags, and phone numbers. These types of data would no longer need to be read by a crowdsource worker or human labeler in order to generate labels for training. More importantly, our method is also independent of LFs trained on the same data as the domain. For e.g, SWEET excluding LFs trained on HTGEN and HTUNSUP (SWEET − *DomainData*) performs as well or even better than SWEET.

## 7 Conclusion

In this paper, we introduced SWEET, a weak supervision pipeline that extracts person names from escort ads without the need for manual labeling. The experimental evaluations show that SWEET achieves SOTA results on the HTNAME dataset with an F1 score of 0.87, outperforming the previous SOTA by 9% while also generalizing better to CoNLL2003 and WNUT2017 datasets. Moreover,

SWEET maintains or improves this performance on removing any LFs trained on relevant domain data. SWEET does not require any human annotators and labeling which is a very important improvement over previous methods because it allows it to be applied easily to real-world datasets. It can also be easily adapted to other domains by designing specific new labeling functions. We also release a new public escort ad dataset HTGEN that uses ChatGPT for both data generation and labeling, which will improve reproducibility and benchmarking in this domain where most data cannot be shared publicly, as it contains personal information.

## 8 Limitations

Despite SWEET having achieved SOTA results on HTNAME, several limitations need to be considered while interpreting the findings.

1. **Dataset Size and Diversity**: The HTNAME database used in this research is a modified version of HT1k from Li et al. (2022). The dataset contains a limited number of data (around 1000 ads). 5-fold evaluation is deployed to accommodate for this small size. While generating HTGEN, although efforts were made to ensure its representativeness and diversity, HTGen's data may still not be 100% representative of real data. This may limit the generalizability of the models trained on it to other domains or datasets.

2. **Explainability and transparency**: HT is a high-stakes application domain and transparency, explainability, and accuracy are all equally important. SWEET has the ability to aggregate multiple models together to get a final

prediction. However, it uses HMM for aggregation which is more complex and less explainable than the ad-hoc way deployed by Li et al. (2022). Moreover, the output of the fine-tuned LFs is also less interpretable. SWEET sacrifices some explainability for increased performance. However, thanks to its modularity, the HMM aggregator can be swapped out for a more explainable method.

3. **Cost**: We were able to improve performance by aggregating BERT models fine-tuned on a variety of NER datasets. This is time-consuming and costly. We used industrial-grade GPUs (RTX8000, V100 and A100) to fine-tune the BERT models.

4. **Privacy**: The HT domain contains private information like a person's name, address, social media tag, email, and phone number. Although the HTNAME and HTUNSUP datasets consists of ads scraped from public websites, these websites may not be fully regulated, and thus, data coming from these ads may violate the privacy of individuals being advertised in those ads. We had to refrain from releasing these datasets because of privacy and ethical concerns. Thus, directly replicating our results on HTNAME and HTUNSUP is not possible. To help counteract this and improve reproducibility, we released HTGEN.

5. **Lack of variety**: We have shown that the HTGEN dataset generated by ChatGPT contains realistic-sounding ads. However, the ads tend to be more repetitive than our datasets (HTUNSUP, HTNAME) with real data scraped from the web. Changing the temperature settings and other parameters showed promise at boosting the variation in the generations but the model was more inclined to produce nonsensical sentences. Future work should aim at increasing variety in the generated ads and keep noise withing a desired range. With more variety of data in the generations, it would also be possible to decrease the size of HTGEN by reducing the number of redundant examples.

## Ethics Statement

Rigorous ethical considerations were taken during the process of this research. The data we use is publicly available, and due to the nature of the advertisements, there are no reasonable expectations for privacy. However, due to the sensitivity of the data, Ethics Approval has been obtained from the Research Ethics Board Office at the authors' university for using this type of data. We have also studied the current best practices for the project through a commissioned Responsible AI Institute evaluation, one of whose recommendation was to focus on a human-centered design. To this end, we have biweekly consultations with human trafficking survivors and have been mindful of not reproducing biases in the design of the algorithm. No personal attributes such as age, physical descriptors, ethnicity, etc were used. In addition, we have also reviewed the current law and policy implications through a comprehensive legal risk assessment and mitigation memorandum from a law firm. As the laws and policies in this domain are catching up with the technologies, we also plan to conduct research in close collaboration with domain experts into the most ethical approach to AI in this domain. Lastly, for transparency and accountability, all of the algorithms being developed will be made available online. Data scraping scripts and our datasets (except those synthetically generated) will not be shared publicly to maintain confidentiality and anonymity.

## 9 Acknowledgements

This research is partially funded by the Canada CIFAR AI Chairs Program and Samsung-Mila Research Grant.

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

**Reader's discretion: Appendix D contains examples of escort ads that contain vulgar and explicit language. If you prefer to avoid explicit language or find such content uncomfortable, we advise you to stop reading before reaching appendix D.**

## A  Labeling Functions

**Fine-tuned Models**  For fine-tuning the *Token-Classification* models, we use a wrapper from Huggingface (Wolf et al., 2020) *AutoModelForToken-Classification* class to compile different backbones, here RoBERTa and DeBERTaV3. We adopt the train-test split of CONLL2003 and WNUT2017 from the Huggingface datasets (Lhoest et al., 2021) for the convenience of reproducing the results ("*wnut_17*" for WNUT2017, "*conll2003*" for CoNLL2003). For FewNERD-L1 and WikiNER-en, we directly downloaded the official stored file and parsed it to the same format as a Dataset loaded from the Datasets Library for running the experiments.

In the training stage, the learning rate is relative to batch size, which is $2 \times 10^{-5} \times (batch\_size/128)$. We train for 5 epochs with AdamW optimizer (Loshchilov and Hutter, 2017) (weight_decay=0.01) on one A100 (40GB) GPU. Detailed information about both the size of the training datasets and the used training batch size are listed in Table 1.

**Individual Labeling Functions**  We provide the performance of individual labeling functions in this section. Table 7 shows the performance of LFs used in the NEAT (Weakly Supervised) experiment in Table 3. Table 8 and Table 9 shows the results of individual fine-tuned models used as LFs in SWEET.

| Model | F1 | Precision | Recall |
|---|---|---|---|
| Label-Indep NEAT Rules | .43 ± .03 | .88 ± .02 | .28 ± .02 |
| Label-Indep NEAT Dictionary | .78 ± .02 | .83 ± .02 | .74 ± .02 |
| Label-Indep NEAT Disambiguation | .79 ± .02 | .80 ± .02 | .77 ± .02 |

Table 7: NEAT-based LFs scores on HTNAME evaluated on 5-fold test sets.

| Model | Dataset | F1 | Precision | Recall |
|---|---|---|---|---|
| DeBERTaV3-base | HTUNSUP | .67 ± .11 | .82 ± .14 | .57 ± .14 |
| | CoNLL2003 | .97 ± .02 | **.98 ± .02** | **.99 ± .02** |
| | Few-NERD-L1 | .92 ± .01 | .95 ± .01 | .91 ± .01 |
| | WikiNER-en | .96 ± .01 | .97 ± .02 | .97 ± .02 |
| | WNUT2017 | .58 ± .04 | .56 ± .04 | .65 ± .04 |
| RoBERTa-base | HTUNSUP | .74 ± .06 | .84 ± .07 | .60 ± .07 |
| | CoNLL2003 | **.98 ± .02** | **.98 ± .02** | **.99 ± .02** |
| | Few-NERD-L1 | .92 ± .01 | .94 ± .02 | .92 ± .02 |
| | WikiNER-en | .93 ± .03 | .94 ± .05 | .97 ± .05 |
| | WNUT2017 | .89 ± .02 | .87 ± .02 | .88 ± .02 |

Table 8: Fine-tuned model LFs of SWEET on CoNLL2003 evaluated on 5-fold test sets.

| Model | Dataset | F1 | Precision | Recall |
|---|---|---|---|---|
| DeBERTaV3-base | HTUNSUP | .41 ± .11 | .50 ± .14 | .52 ± .14 |
| | CoNLL2003 | **.72 ± .04** | .71 ± .08 | **.73 ± .08** |
| | Few-NERD-L1 | .66 ± .12 | .82 ± .13 | .41 ± .13 |
| | WikiNER-en | .65 ± .11 | .69 ± .13 | .52 ± .13 |
| | WNUT2017 | .62 ± .09 | .71 ± .15 | .47 ± .15 |
| RoDERTa-base | HTUNSUP | .53 ± .07 | .56 ± .13 | .63 ± .13 |
| | CoNLL2003 | .71 ± .05 | .67 ± .08 | .71 ± .08 |
| | Few-NERD-L1 | .61 ± .10 | .67 ± .10 | .45 ± .10 |
| | WikiNER-en | .65 ± .10 | .63 ± .09 | .57 ± .09 |
| | WNUT2017 | .67 ± .09 | **.81 ± .14** | .48 ± .14 |

Table 9: Fine-tuned model LFs of SWEET on WNUT2017 evaluated on 5-fold test sets.

## B  ChatGPT and the HT Domain

ChatGPT is an advanced language model developed by OpenAI and built upon the GPT architecture. Although the training data that is used to train ChatGPT is not made public, it is built upon the GPT (Generative Pre-trained Transformer). These models were trained on Common Crawl, a dataset that includes a wide range of web content and ad-related content. By providing essential background and convincing ChatGPT that it is a researcher that studies human trafficking, we were able to guide it to generate a list of patterns generally seen in escort ads (Figure 2).

1. "Beautiful roses for sale - come and pick your favorite bouquet today" - this ad uses the rose symbol to lure potential victims into a false sense of security, before offering them up for sale.

2. "Outcall girls available now - come experience the pleasure of our talented escorts" - this ad uses the term "outcall" to suggest that the trafficked individuals will be brought to the customer, rather than the customer going to them.

3. "Meet our lovely ladies - Lisa, Kate, and Lexi are waiting for you to call" - this ad uses nicknames instead of real names, which is a common tactic used by traffickers to avoid detection.

4. "Grilfriend experiens - call now for a good time with the hottest girls in town" - this ad employs misspellings to evade filters, as well as intimating that these girls are offering a kind of relationship experience in order to appeal to vulnerable individuals.

By finding and analyzing similar ads to these, researchers will be able to develop a better understanding of the patterns and language used by human traffickers, ultimately enabling law enforcement to more effectively combat this heinous activity.

Figure 2: ChatGPT's Understanding of HT Domain

## C  Problems with LLMs

Large language models, such as ChatGPT have garnered recognition from researchers for their versatility on a wide range of NLP tasks while exhibiting commendable performance compared to other SOTA models. Nevertheless, it is imperative to acknowledge that these models have also faced significant scrutiny and critical appraisals. In this context, we mainly highlight the issue of *Encoding Bias*.

Due to the large size of the datasets used to train LLMs, it is very difficult to check and verify every part of the training dataset for potential bias. Thus, it is common for large language models to exhibit various types of bias as stated in (Bender et al., 2021; Hutchinson et al., 2020; Kurita et al., 2019). When building HTGEN, we observed that the generated text had a propensity to include derogatory language (Figure 3), and sexually explicit words (Figure 4). It also generated ads that have a negative sentiment towards the black community (Figure 5). Of all the examples generated in HTGEN, black people are denied services more than any other ethnic group. No mention of the words found in Figures 3 and 4 had occurred in any prompting processes. This ability to generate realistic sounding ads suggests that there is a good chance it had exposure to escort ads during its training process.

## D  Examples of escort ads

**Sexy Cougar - 41** Hello there, my name is Gina. I'm a sexy and mature brunette with curves in all the right places. Let me show you what a real woman can do. I'm experienced and know how to please a man. Incalls only. Call or text me at 905-XXX-XXXX to book an appointment.

Figure 3: Derogatory or Offensive Language

**SeXy BlAcK MiLF**🌀💦🌺 Have you ever tried a Sexy, spicy Black woman? Look no further, I'm the perfect combination of Sexiness, Beauty, Intelligence, and personality. I'm Angell, 38 years old and I'm the perfect woman for you. I have smooth chocolate skin, an amazing body and I know how to please. I'm completely independent and discreet, always ready to satisfy all your needs. I provide all safe services, no rushed sessions, and easy parking. Call or text me now at (XXX)-XXX-XXXX. No AA or Trades Please.

Figure 4: Usage of Vulgar and Sexually Explicit Words

Young SPANISH BEAUTY IN TOWN XoXo Outcalls 24/7 Hi! I am a young, sexy, and beautiful Spanish girl ready to provide you with a mesmerizing experience. I offer outcalls 24/7, and I am always discreet and professional. I provide an unforgettable experience and satisfy all your fantasies. Call or text me to book an appointment. No black men. Xoxo

Figure 5: Racism in HTGEN

Title: 💋💦Michelle's $80 CARCALL Special!!!💦💋 Description: LOWBALLERS WILL BE BLOCKED!!! $80 BBB---J $130 F--S $160 BB--FS 123-456-789💋

Figure 6: Example ad From HTNAME

Title: GisellaGorgeous Puerto Rican😍😘😍 Description: My name is Gisella sexy little petite thing with curves that don't stop 💕 I am excited to be your grown up bestie and come over for a private playdate. I love to make new friends! 🍭🎀🎀 Let's get together and see what kind of trouble we can stir up!😘😋😘 (916)888-7777 no extra photos don't ask for any, I am not a cheap commodity🚫 incalls only.

Figure 7: Example ad From HTGEN