# OpenReview forum: "SWEET - Weakly Supervised Person Name Extraction for Fighting Human Trafficking"
_EMNLP/2023/Conference — EMNLP 2023 Findings_

### Official Review · Reviewer_6zce · 2023-07-30

**Soundness:** 3

**Excitement:**

3: Ambivalent: It has merits (e.g., it reports state-of-the-art results, the idea is nice), but there are key weaknesses (e.g., it describes incremental work), and it can significantly benefit from another round of revision. However, I won't object to accepting it if my co-reviewers champion it.

**Paper Topic And Main Contributions:**

The topic of this paper is to extract person name entities from escort advertisements using weak supervision methods. The paper introduces the SWEET method to address the problem and with up to a 9% increase in F1 score compared to the current SOTA method on benchmark datasets.To facilitate further research in this domain, the paper also introduces the HTGEN dataset, which is generated by having ChatGPT synthesize synthetic escort ads. Unlike real-world datasets, HTGEN is shareable, contributing to research reproducibility and sharing.

**Questions For The Authors:**

1.Since ChatGPT is capable of achieving optimal results on tasks and can effectively explain prediction outcomes, could this influence the further research value of the task?

2.In the paper, the authors mentioned using the Skweak framework for weakly supervised learning, but they did not elaborate on why they chose the Skweak framework. I would like to know the advantages of Skweak compared to other weakly supervised frameworks and how it contributes to improving the performance of the SWEET method.

**Reasons To Accept:**

1. Introducing a novel solution: SWEET, as a weak supervision method, offers a fresh approach to the task of extracting person names from escort ads. Its efficiency and effectiveness surpass existing methods.

2. Providing a shareable dataset: The development of the HTGEN dataset offers a collection of synthetic escort ad data that can be shared with the research community.


**Reasons To Reject:**

1. The methods mentioned in the paper mainly involve aggregating existing approaches, lacking originality, and there is no clear explanation of why these methods can address data noise issues in Person Name Extraction.

2. The paper only tested ChatGPT on the HTName dataset and did not evaluate it on other datasets. However, ChatGPT's performance on the HTName dataset outperformed the proposed method in the paper, indicating that the proposed method is not state-of-the-art (SOTA).

3. The paper used relatively few of the latest baselines, and compared to these baselines, the improvements brought by the proposed method in the paper are limited.

4. Table 3 is missing the evaluation results of SWEET on the HTGen dataset.

**Reproducibility:**

3: Could reproduce the results with some difficulty. The settings of parameters are underspecified or subjectively determined; the training/evaluation data are not widely available.

**Reviewer Confidence:**

5: Positive that my evaluation is correct. I read the paper very carefully and I am very familiar with related work.

---

> ### Author Rebuttal · Authors · 2023-08-29
>
> We thank the reviewer for their remarks and hope to answer their questions below:
>
> **Q1. The methods mentioned in the paper mainly involve aggregating existing approaches, lacking originality, and there is no clear explanation of why these methods can address data noise issues in Person Name Extraction.**
>
> The noisy nature of the data demonstrates the challenges related to annotation and explains why the existing approaches underperform. In our real-world HT domain, where it is hard to access labeled data and the text is noisy and unstructured, we show that treating the labels from existing approaches and aggregating them in a weak supervision framework can achieve superior and robust performance, as our experiments confirm this.
>
> In particular, we provide an interesting and novel ensemble approach of combining multiple classifiers that leverages 6 different datasets (ConLL, WNUT2017, Few-NERD-LI, WikiNER-en, HTUnsup, and HTGen) while providing both improved results and the ability to work in a transfer-learning setup.
>
> By leveraging multiple signals our model can easily adapt to different types of data which contributes to the high performance on both the HT domain data and general settings. Moreover, we present a way to generate noisy data that matches the domain and is labeled automatically with ChatGPT, to achieve better performance on the in-domain HTName dataset. We will highlight the novelty better in the revised version of the paper.
>
>
> **Q2. The paper only tested ChatGPT on the HTName dataset and did not evaluate it on other datasets. However, ChatGPT’s performance on the HTName dataset outperformed the proposed method in the paper, indicating that the proposed method is not state-of-the-art (SOTA).**
>
> Although ChatGPT is the best performing model, there are concerns regarding its use on escort ads. First, there is some financial **cost** associated with employing ChatGPT on large datasets of escort ads which limits its accessibility whereas our method is open-source and completely free to use. Second, the **stability** of the OpenAI API output is a concern as discussed in [8]. We have found that even the performance of the 0613 versioned GPT-3.5 endpoint can vary and result in performance worse than BERT and our method. While this does not hurt our method much where one only needs to get good performance a single time (to provide weak labels on a fixed dataset), and one could consider even the more powerful GPT-4, it means it would be unreliable to build a system for real-world applications based on ChatGPT. Finally, depending on the method used to access ChatGPT, there can be **privacy** concerns (for example, it can be used in HIPAA compliant way - i.e., to the standard of US health data privacy regulation - by accessing it through Azure, but that requires approval, and the standard OpenAI endpoint is not HIPAA compliant) with this sensitive data. The intended use of our method is on large scale real-time ads with sensitive information, and using ChatGPT, is not feasible due to cost, stability and privacy issues mentioned above. We appreciate the reviewer’s comment and will edit our discussion section to better include these points in the revised version.
>
> **Q3. The paper used relatively few of the latest baselines, and compared to these baselines, the improvements brought by the proposed method in the paper are limited.**
>
> We respectfully disagree and highlight that SWEET gains a 9% improvement on F1-score over baselines on the main domain dataset - HTName. Our method beats NEAT [1], the previous SOTA in this domain and hence we excluded some of the baselines from [1]. Additionally, the performance of individual fine-tuned LMs which are used as LFs are provided in Table 4 which also act as recent baselines where SWEET’s F1-score (0.87) is higher than the best performing LF (0.82).
>
> **Q4. Table 3 is missing the evaluation results of SWEET on the HTGen dataset.**
>
> SWEET leverages LFs which includes roberta and deberta models fine-tuned on both general NER datasets and domain-specific HTGen. Hence, evaluating our method on HTGen would not be a fair comparison given our method has seen all the HTGEN data. However, in Table 6, we provide the results of *SWEET - HTGen LFs* on HTGen (0.97 F1) which performs on par with NEAT in Table 3 but with higher precision. In the next version of the paper, we plan to add a small test set for HTGen, which is not used for fine-tuning the labeling functions in SWEET and have that entry filled in Table 3, to reduce confusion. Thank you for your feedback.
>
> **Q5.Since ChatGPT is capable of achieving optimal results on tasks and can effectively explain prediction outcomes, could this influence the further research value of the task?**
>
> We thank the reviewer for bringing up this interesting discussion point.
>
> With the advent of ChatGPT, we believe that this question may be raised for several NLP tasks. For the task at hand, some of the concerns regarding the usage of ChatGPT have been addressed in our response to the above question 2. Moreover, for the given task, we believe that a model like ChatGPT is an overkill and we propose leveraging smaller LLMs with much fewer parameters (and training cost and computational requirements) and show that it provides competitive results. NER results are also easily explainable as opposed to more complicated tasks which may benefit from the explanations that ChatGPT is able to provide, for example, misinformation detection.
> Lastly, there are studies [4] investigating ethical concerns and challenges of using ChatGPT which need to be further understood before relying on it completely for solving similar real-world problems.
>
> **Q6. In the paper, the authors mentioned using the Skweak framework for weakly supervised learning, but they did not elaborate on why they chose the Skweak framework. I would like to know the advantages of Skweak compared to other weakly supervised frameworks and how it contributes to improving the performance of the SWEET method.**
>
> Snorkel[2] and Skweak [3] are the two most popular frameworks for programmatic weak supervision. Skweak (2021) is a newer toolkit than Snorkel (2017) and was specifically designed for NLP tasks. Thus it is more fitting for our HT NER task. Skweak was also built using SpaCy, which facilitates downstream processing. Skweak’s support for Spacy also made it easier to convert the NEAT components to labeling functions for the *NEAT (Weakly Supervised)* experiment mentioned in Table 3. Moreover, Skweak provides a variety of different labeling function types: heuristics, Gazetteers and Document-level functions which allow us to combine weak signals from both rule based systems and LLMs. We will include this explanation in the revised version of the paper.
>
> **References:**
>
> [1] Li, Yifei, et al. "Extracting Person Names from User Generated Text: Named-Entity Recognition for Combating Human Trafficking." Findings of the Association for Computational Linguistics: ACL 2022. 2022.
>
> [2] Ratner, Alexander J., et al. "Snorkel: Fast training set generation for information extraction." Proceedings of the 2017 ACM international conference on management of data. 2017.
>
> [3] Pierre Lison, Jeremy Barnes and Aliaksandr Hubin (2021), "skweak: Weak Supervision Made Easy for NLP", ACL 2021 (System demonstrations).
>
> [4] Zhou, Jianlong, et al. "Ethical ChatGPT: Concerns, challenges, and commandments." arXiv preprint arXiv:2305.10646 (2023).
>
> [5] Veselovsky, Veniamin, Manoel Horta Ribeiro, and Robert West. "Artificial Artificial Artificial Intelligence: Crowd Workers Widely Use Large Language Models for Text Production Tasks." arXiv preprint arXiv:2306.07899 (2023).
>
> [6] Fabrizio Gilardi, Meysam Alizadeh, & Maël Kubli (2023). ChatGPT outperforms crowd workers for text-annotation tasks. Proceedings of the National Academy of Sciences, 120(30), e2305016120.
>
> [7] U.N. (2021). U.N News. Traffickers abusing online technology, UN crime prevention agency warns.
>
> [8] Chen, Lingjiao, Matei Zaharia, and James Zou. "How is ChatGPT's behavior changing over time?." arXiv preprint arXiv:2307.09009 (2023).

---

### Official Review · Reviewer_pn8R · 2023-07-31

**Typos Grammar Style And Presentation Improvements:** No
**Soundness:** 3

**Excitement:**

3: Ambivalent: It has merits (e.g., it reports state-of-the-art results, the idea is nice), but there are key weaknesses (e.g., it describes incremental work), and it can significantly benefit from another round of revision. However, I won't object to accepting it if my co-reviewers champion it.

**Missing References:**

No

**Paper Topic And Main Contributions:**

To address the human trafficking problem, this paper proposes a weakly supervised method to extract person names from noisy texts. Specifically, utilize heuristic rules to filter out false positive entities, and employ models tuned on existing NER datasets for silver data generation. This silver data will then be used for training on human trafficking data at a later stage.

**Questions For The Authors:**

No

**Reasons To Accept:**

1. This work has some novelty: transferring the knowledge from exisiting NER models/datasets for noisy text NER tasks.

2. The results seem good.

**Reasons To Reject:**

1. The current setting, which involves using the model (fine-tuned roberta/deberta) trained on HTUNSUP and HTGEN to indirectly teach another model (HMM), should not be referred to as weak supervision. If you have access to the train sets of HTUNSUP and HTGEN, it would be more performant to directly train a model on them or compare your HMM method with other baselines that fully utilize these training sets. Essentially, your method is no different from these fully-supervised baselines since both have access to the full training set but  you use it in an indirect way deliberately.

2. The writing can be largely improved and I can't follow well.

3. Admittedly, human trafficking is an important problem but I don’t see how extracting person names can help largely help. I personally think NER on noisy text is a pretty general and important task so I would love to see this method be applied to general domains(in the correct setting).

4. If the motivation is to address the NER problems in noisy texts, then maybe performing NER on truly noisy texts is a better idea instead of testing the methods on CoNLL2003 and WNUT2017. Testing a model that has been indirectly trained using a training set is not considered generalization.

**Reproducibility:**

4: Could mostly reproduce the results, but there may be some variation because of sample variance or minor variations in their interpretation of the protocol or method.

**Reviewer Confidence:**

4: Quite sure. I tried to check the important points carefully. It's unlikely, though conceivable, that I missed something that should affect my ratings.

---

> ### Author Rebuttal · Authors · 2023-08-29
>
> We thank the reviewer for their remarks and hope to answer their questions below:
>
> **Q1. The current setting, which involves using the model (fine-tuned roberta/deberta) trained on HTUNSUP and HTGEN to indirectly teach another model (HMM), should not be referred to as weak supervision. IF you have access to the train sets of HTUNSUP and HTGEN, it would be more performant to directly train a model on them or compare your HMM method with other baselines that fully utilize these training. Essentially, your method is no different from these fully-supervised baselines since both have access to the full training set but you use it in an indirect way deliberately.**
>
> Our focus in this paper is name extraction from escort ads for combatting HT. For HTName and HTGEN datasets, we do not use true labels for training the labeling functions (fine-tuned models). We will improve the writeup in the paper to better clarify that we do not have access to in-domain true labels. This is why we call it weakly supervised training. The labels used to fine-tune roberta/deberta on HTUNSUP and HTGEN are weak labels obtained from ChatGPT. The performance of only using those trained models for our task is reported in Table 4 (e.g. 0.67 for deberta on HTUnsup), which is significantly lower than what we achieve with SWEET (0.87). NEAT, the previously state-of-the-art model in this domain, does depend on a training set of HT data directly from the main target dataset HTName. We compare against the previously best model in such a setting, and show our approach gives better performance, even in this “unfair” comparison setting where NEAT gets to use in-distribution labeled training data and our method does not.
>
> In table 5, second row, we report the performance of SWEET without having access to any domain data, i.e, performance on HTName without having the labeling functions that were fine-tuned with ChatGPT-labeled HT data. Similarly the results for CoNLL and WNUT are noted after removing the labeling functions that were fine tuned on the training split of the CoNLL and WNUT datasets respectively. We can see that SWEET without domain data still performs as good (or even better in case of WNUT), compared to when it has the target domain labeling functions (SWEET in table 3).
>
> **Q2. Admittedly, human trafficking is an important problem but I don’t see how extracting person names can help largely help. I personally think NER on noisy text is a pretty general and important task so I would love to see this method be applied to general domains(in the correct setting).**
>
> This paper presents an approach for NER for HT, which is a part of our bigger project on finding HT in escort ads by looking for organized activity which advertises multiple people. A majority of traffickers have multiple victims, and hence one major indication of HT is the presence of multiple people being advertised within the same ad or similar groups of ads[1, 7]. If an ad or a group of connected ads advertise multiple names, it is likely that the activity is related to trafficking. This is considered along with other posting patterns (geographical movements, temporal burstiness, etc.). Extracting the advertised names from these ads is critical to scalably flag such suspicious ads for further investigation. We agree that NER on noisy text in general is an important task but it is not our current focus. The focus of this work is to show how LFs trained on non-domain data can be better leveraged to efficiently extract names from noisy escort ads.
>
> **Q3. If the motivation is to address the NER problems in noisy texts, then maybe performing NER on truly noisy texts is a better idea instead of testing the methods on CoNLL2003 and WNUT 2017. Testing a model that has been indirectly trained using a training set is not considered generalization.**
>
> The main goal of this paper is improving NER performance for the HT text which contains data that is sensitive and hard to manually label. We report the performance on CoNLL and WNUT to show our proposed method, unlike NEAT which is the previous state-of-the-art for NER on HT data, generalizes well to these commonly used benchmark datasets. This, along with the HTGen, helps with reproducibility of the results. We are extending our work to include more datasets used in noisy NER methods
>
> We will improve the writing to better reflect these points in the revised version.
>
> **References:**
>
> [1] Li, Yifei, et al. "Extracting Person Names from User Generated Text: Named-Entity Recognition for Combating Human Trafficking." Findings of the Association for Computational Linguistics: ACL 2022. 2022.
>
> [2] Ratner, Alexander J., et al. "Snorkel: Fast training set generation for information extraction." Proceedings of the 2017 ACM international conference on management of data. 2017.
>
> [3] Pierre Lison, Jeremy Barnes and Aliaksandr Hubin (2021), "skweak: Weak Supervision Made Easy for NLP", ACL 2021 (System demonstrations).
>
> [4] Zhou, Jianlong, et al. "Ethical ChatGPT: Concerns, challenges, and commandments." arXiv preprint arXiv:2305.10646 (2023).
>
> [5] Veselovsky, Veniamin, Manoel Horta Ribeiro, and Robert West. "Artificial Artificial Artificial Intelligence: Crowd Workers Widely Use Large Language Models for Text Production Tasks." arXiv preprint arXiv:2306.07899 (2023).
>
> [6] Fabrizio Gilardi, Meysam Alizadeh, & Maël Kubli (2023). ChatGPT outperforms crowd workers for text-annotation tasks. Proceedings of the National Academy of Sciences, 120(30), e2305016120.
>
> [7] U.N. (2021). U.N News. Traffickers abusing online technology, UN crime prevention agency warns.
>
> [8] Chen, Lingjiao, Matei Zaharia, and James Zou. "How is ChatGPT's behavior changing over time?." arXiv preprint arXiv:2307.09009 (2023).

---

### Official Review · Reviewer_5iZ8 · 2023-08-04

**Typos Grammar Style And Presentation Improvements:** Please see comments above.
**Soundness:** 3

**Excitement:**

4: Strong: This paper deepens the understanding of some phenomenon or lowers the barriers to an existing research direction.

**Paper Topic And Main Contributions:**

The paper proposes an adaptation of weakly supervised system of noisy text NER. Specifically it develops labeling functions and heuristic rules for NER from human trafficking posts. The paper is well drafted with relevant details for anyone with little knowledge of weakly supervised NLP. The paper besides provides extensive evaluation of the pipeline showing various benefits and drawbacks of the components. Meanwhile the limitation section is the key highlight, where the authors effectively justify and accept the weakness of the said pipeline which is rarely seen in research papers.  Overall a solid empirical paper!!

**Questions For The Authors:**

A. Though there are few works on weakly supervised NER for Noisy text, there are many for weakly supervised NER. Those are more ideal benchmarks than methods like spacy NER and flair? Was there are any specific reason behind selection of these over weakly supervised NER for effective benchmarking.
B. The ratio of data from HT's (unsup + Gen) to HTName is very low, with HTName having significantly lesser samples. What was the reason behind not looking at improving the test set? For example, synthetic test set with human evaluation would be more helpful in moving the area forward?

**Reasons To Accept:**

In addition to contributions mentioned above, following are few reasons for acceptance

- The paper is well drafted and easy to follow
- The literature selected and area of focus though narrow has large value in HT post mining and NLP community.
- Moreover the authors do introduce new synthetic dataset, which was effectively analyzed in this work.
- The methods selected are well established making the study suitable for fostering further research in this areas.
- The empirical results are exhaustive enough to justify the results.


**Reasons To Reject:**

The paper argues as weak supervision for NER from noisy Human trafficking post, but the benchmark is very weak, infact uses only 995 samples from the domain. Meanwhile the authors focus on showing generalizability against CoNNLL, WNUT which begs to question if the paper focus is on NER generalization or NER for Human trafficking post. Besides the total number of named entities etc are completely unknown for HTName which is used for testing. Moreover some of the results in Table 5, show that simple LF's with source domain data to perform onpar with SWEET, which again is another area of concern. Moreover weakly supervised NER framework is compared with regular NER methods for justification (other than NEAT).


Following the key points to be improved for the paper

- Lines 005 - 007 regarding combining rule matching with generalization for weak labels is unclear. Rather you can mention the specific labelling function capability of the sweet architecture here in brief.
- Line 017 - 018 why create synthetic data when there are existing SoTA? Please add some details on the same to keep the work interesting to audience. For example lack of public sets (if it is the reason?)
- Line 090-095, can you characterize efficiency quantitatively? For example, you can say something like it takes 50% less time to build model with proposed approach (if something like this is indeed possible)
- Line 082-084: How does the proposed idea fair against https://dl.acm.org/doi/abs/10.1145/3502223.3502228 and other similar works, especially in terms of novelty. Because reading sections 3 shows the LF's used are fairly heuristical and common. Meanwhile the 6 models used are meant for regular NER. If there are models that are specific to noisy text it is more better to cross compare on the results. Often because of the fact that the noisy data results heavily depend on how the noise is modeled.
- Line 265 - Please mention on the rationale behind selection such diverse sets, especially non noisy dataset like CONNLL
- Line 391 - Till here most of the sections are clearly written, however there is one question which is unanswered. The definition of noise which is being addressed. Specifically how is the variance of noise in HK dataset, otherwise one can argue that LF's are more overfit for the data in hand. Besides many of these posts exhibit syntactic patterns. So in the explanation of datasets used for evaluation and building LF's such details need to be precisely presented to ground the arguments in section 5.
Besides the test set of HTName is very small, I wonder if the results are  debatable?
- Line 468 - What is the purpose of evaluation on CoNLL 2003, especially the main argument of the entire paper is on noisy text. Meanwhile from results we can see that Antirules are acting like filters leading to drop in results, which supports my earlier argument on line 082. Please reconsider this section to accomodate such details.
Meanwhile Table 5 and 6 are hard to follow and correlate with earlier sections. For example, please define  SWEET Include like roberta + deberta with many datasets + frequency rules. Where both LMs are trained on various XYZ dataset (From section 4). Then add that - HTGEN LF's include models of roberta deberta with HTGen data only. for comparison. Not the others. Because comparing SWEET which has many LF's + other components with only LF's of specific type needs to be clearly highlighted with rationale, else it would reflect as benefit of other components. For example, The takeaway from Table 5, is to use HTGen LF's + Antirules but not full sweet system? Because in Table 6, we can see removing these the results are not as high.
Line 509 - Entire section has no value add, rather please merge this with limitation sections to justify benefit of work.




**Reproducibility:**

2: Would be hard pressed to reproduce the results. The contribution depends on data that are simply not available outside the author's institution or consortium; not enough details are provided.

**Reviewer Confidence:**

4: Quite sure. I tried to check the important points carefully. It's unlikely, though conceivable, that I missed something that should affect my ratings.

---

> ### Author Rebuttal · Authors · 2023-08-29
>
> We thank the reviewer for their remarks and hope to answer their questions below:
>
> **Q1. The paper argues as weak supervision for NER from noisy Human trafficking post, but the benchmark is very weak, infact uses only 995 samples from the domain. Meanwhile the authors focus on showing generalizability against CoNLL, WNUT which begs to question if the paper focus is on NER generalization or NER for Human trafficking post. Besides the total number of named entities etc are completely unknow for HTName which is used for testing. Moreover some of the results in Table 5, show that simple LF’s with source domain data to perform onpar with SWEET, which again is another area of concern. Moreover weakly supervised NER framework is compared with regular NER methods for justification (other than NEAT).**
>
>
> Due to the explicit and sensitive nature of escort ads (which is the main source of data in HT detection studies), there are no publicly available datasets particularly for the task of extracting person names from these ads. The only known private dataset is HTName from previous work [1] which is small and contains only 995 samples. Table 1 from [1] provides insights on the number of names, etc in HTName and we will include this information in our camera-ready version, if accepted.
>
> The previous SOTA for the HT domain, NEAT, had poor performance on CoNLL and WNUT. So in our work, we show that our method outperforms it on both the domain-dataset (HTName) as well as general NER datasets (CoNLL and WNUT). To clarify, our motivation for including experiments on these general datasets is not for NER generalization but rather to show that SWEET is a very flexible pipeline that can be adjusted based on the task at hand. LFs can be easily swapped out and exchanged with other LFs.
>
> On table 5, it is true that the simple LFs with source domain data perform on par with SWEET on HTName, however they do not generalize well on other datasets and given the small size of HTName and low costs of adding more labeling functions, one can expect SWEET to perform better in the real-world.
>
> We will revise our writing in the paper to better reflect these points.
>
> **Q2.Though there are few works on weakly supervised NER for Noisy text, there are many for weakly supervised NER. Those are more ideal benchmarks than methods like spacy NER and flair? Was there any specific reason behind selection of these over weakly supervised NER for effective benchmarking.**
>
> Within the weak supervision paradigm, our work falls under programmatic weak supervision where instead of 1 weak label, each data point is associated with multiple weak labels which are aggregated using the HMM. In this way, it is inherently different from weakly supervised methods that deal with a single weak label or label noise. To compare with the weakly supervised NER baselines, we would need to pick one of the LFs to act as the “weak label” which makes it a slightly different problem setup than the one we focus on. Our experiments aim to show that aggregating multiple weak labels provides better results compared to using pre-trained methods like Flair, Spacy or other transformer-based models.
>
>
> **Q3. The ratio of data from HT's (unsup + Gen) to HTName is very low, with HTName having significantly lesser samples. What was the reason behind not looking at improving the test set? For example, synthetic test set with human evaluation would be more helpful in moving the area forward? why create synthetic data when there are existing SoTA?**
>
> HTName is a relatively small dataset consisting of 1000 ads used in the paper of one of our benchmarks [1] (referred to as HT1k in that paper). Since then we introduced HTUnsup which is 6 times the size of the HTName]. HTName was labeled using Mechanical Turk and there are recent works that show that these workers are increasingly relying on ChatGPT [5] and other recent works that discuss how ChatGPT does better than the human labellers for annotations [6].
>
> Aside from that, the data naturally contains explicit language and sensitive information like email, phone number, and address and thus using manual labellers is not ideal. This difficulty in labeling this data is one of the main motivating factors for a weakly supervised solution. Moreover, synthetic labeling is significantly more cost effective. In view of these considerations, a synthetic dataset with synthetic labellers allows for more privacy and allows us to scale up more rapidly and at a cheaper cost (as mentioned in section 4.5).
>
>
> **Q4. How does the proposed idea fair against https://dl.acm.org/doi/abs/10.1145/3502223.3502228 and other similar works, especially in terms of novelty. Because reading sections 3 shows the LF's used are fairly heuristical and common. Meanwhile the 6 models used are meant for regular NER. If there are models that are specific to noisy text it is more better to cross compare on the results. Often because of the fact that the noisy data results heavily depend on how the noise is modeled.**
>
> The work cited above uses a similar approach to weak labels as us where a fine-tuned BERT model is used to generate pseudo labels. In our work, however, we have focused on leveraging simple heuristics and predictions from several such fine-tuned language models and aggregating them to make an ensemble. We consider this problem setting to be different from one where a single weak or noisy label is handled and hence did not compare with such methods in our experiments. The cited work is also semi-supervised and is dependent on a small set of labeled samples whereas our method doesn't rely on any manual labels. To provide better comparison with methods for noisy NER, we are extending our work to include more datasets used in noisy NER methods, so that SWEET’s performance can be indirectly compared with them.
>
> **Q5. how is the variance of noise in HK dataset, otherwise one can argue that LF's are more overfit for the data in hand. Besides many of these posts exhibit syntactic patterns. So in the explanation of datasets used for evaluation and building LF's such details need to be precisely presented to ground the arguments in section 5. Besides the test set of HTName is very small, I wonder if the results are debatable?**
>
> We will improve the writing in the revised version to include more details relevant for LF building.
> To clarify, the noisy nature of the data is emphasized to demonstrate the challenges related to its annotation and why simple rule-based approaches and gazetteers alone may fail. None of the LF models were trained on any ground-truth labels in the HT domain and the HMM was trained on only the outputs of the LFs. Hence, overfitting is unlikely. During the training process, HTName remains an unseen test set and the evaluation results provided in Table 3 are the mean and standard deviation over 5-folds.
>
> We acknowledge the remaining comments on specific lines and table results and will include update the paper to reflect them.
>
> **References:**
>
> [1] Li, Yifei, et al. "Extracting Person Names from User Generated Text: Named-Entity Recognition for Combating Human Trafficking." Findings of the Association for Computational Linguistics: ACL 2022. 2022.
>
> [2] Ratner, Alexander J., et al. "Snorkel: Fast training set generation for information extraction." Proceedings of the 2017 ACM international conference on management of data. 2017.
>
> [3] Pierre Lison, Jeremy Barnes and Aliaksandr Hubin (2021), "skweak: Weak Supervision Made Easy for NLP", ACL 2021 (System demonstrations).
>
> [4] Zhou, Jianlong, et al. "Ethical ChatGPT: Concerns, challenges, and commandments." arXiv preprint arXiv:2305.10646 (2023).
>
> [5] Veselovsky, Veniamin, Manoel Horta Ribeiro, and Robert West. "Artificial Artificial Artificial Intelligence: Crowd Workers Widely Use Large Language Models for Text Production Tasks." arXiv preprint arXiv:2306.07899 (2023).
>
> [6] Fabrizio Gilardi, Meysam Alizadeh, & Maël Kubli (2023). ChatGPT outperforms crowd workers for text-annotation tasks. Proceedings of the National Academy of Sciences, 120(30), e2305016120.
>
> [7] U.N. (2021). U.N News. Traffickers abusing online technology, UN crime prevention agency warns.
>
> [8] Chen, Lingjiao, Matei Zaharia, and James Zou. "How is ChatGPT's behavior changing over time?." arXiv preprint arXiv:2307.09009 (2023).

---

### Meta-Review · Area_Chair_nywZ · 2023-09-18

**Recommendation:** 3

**Metareview:**

The paper proposed a new approach for extracting person names from the noisy texts. It used heuristic rules and labeling functions the enhance the labeling results. The experimental results show that the F1 scores are increased by 9% compared to the SOTA methods. The paper is well-organized and easy to follow, except for the writing. The baselines and settings are not very suitable. For example, only texting the performance with ChatGPT on the HTName. Conducting more experiments will make the paper more convincing.

---

### Decision · Program_Chairs · 2023-10-07

**Decision:**

Accept-Findings

**Comment:**

The paper proposed a new approach for extracting person names from the noisy texts. It used heuristic rules and labeling functions the enhance the labeling results. The experimental results show that the F1 scores are increased by 9% compared to the SOTA methods. The paper is well-organized and easy to follow, except for the writing. The baselines and settings are not very suitable. For example, only texting the performance with ChatGPT on the HTName. Conducting more experiments will make the paper more convincing.